# The Mind Under Pressure: What Roles Does Education Play in the Relationship Between Chronic Stress and Cognitive Ability?

**DOI:** 10.3390/jintelligence13020013

**Published:** 2025-01-23

**Authors:** Maximilian Seitz, Diana Steger

**Affiliations:** Leibniz Institute for Educational Trajectories, Wilhelmsplatz 3, 96047 Bamberg, Germany; diana.steger@lifbi.de

**Keywords:** chronic stress, cognitive abilities, education, adulthood

## Abstract

Chronic stress is an important predictor of mental and physical health, but little is known about its association with cognitive abilities and education during the lifespan. We hypothesized that chronic stress would be negatively correlated with cognitive abilities, particularly crystallized intelligence, and that this association would be stronger among individuals with lower educational attainment due to limited stress-coping resources. We used cross-sectional data from the German National Educational Panel Study (NEPS SC6), comprising 10,416 adults aged 29 to 71 years (50.80% female; 49.20% male). Fluid and crystallized intelligence were assessed with a reasoning test and a vocabulary test, respectively; chronic stress was assessed with a questionnaire on social stress and anxiety. The tests and the questionnaire were conceptualized for a heterogeneous and large-scale sample. Our results show small negative associations between chronic stress and both fluid and crystallized cognitive abilities, which persist after controlling for demographic variables. However, there were no significant differences between educational groups. Although the study does not address longitudinal patterns, it highlights the complex interaction between stress and cognition, and it underscores the need for further research to explore how educational resources may mitigate the impact of chronic stress on cognitive health.

## 1. Introduction

In recent years, there has been an increase in chronic stress in many countries ([20]). Likewise, global networks recognize the importance of mental health for individual well-being and a productive society (e.g., [56]). Chronic stress is associated with negative effects on physical health, such as triggering or aggravating diseases of the cardiovascular and gastrointestinal systems, as well as negative effects on mental health, such as depression and anxiety (for a review, see [81]). The negative effects of chronic stress go even further as previous research frequently reported negative associations with basic cognitive abilities (for a review, see [64]). Education is often considered important in this regard, as educational attainment is associated with access to material and cultural resources, opportunities, and privileges important to mental health (for a review, see [36]). However, research on the interrelation between chronic stress, cognitive abilities, and education at different stages in life is limited. Therefore, the current study explores the relationship between chronic stress, cognitive abilities, and education using cross-sectional data from a German panel study featuring adults covering a broad age range.

### 1.1. Chronic Stress

At a basic level, stress can be understood as a psychological or bodily response to cope with the potential threat or loss of homeostasis (i.e., the balance and functioning of the organism) ([22]). It relates to a transactional process of how the organism reacts to real or perceived environmental demands, namely by interpreting them as neutral, positive (e.g., challenges), or negative (e.g., threats) when compared to the individually available coping resources ([49]). Stress is often conceptualized by theoretical frameworks as an interaction of genetic, biological, psychological, and environmental influences ([47]). Acute stress is usually the reaction to short-term stressors that elicit adaptive (i.e., eustress) or maladaptive (i.e., distress) changes in, among others, the nervous and cardiovascular systems, primarily due to the activation of the hypothalamic–pituitary–adrenal axis ([35]; [81]). The biological response to stress is a non-linear sequence of secondary outcomes (e.g., disruptions of the cardiorespiratory and immune systems) influenced by personality traits and behavioral patterns ([35]) associated with, among others, material and cultural resources. However, if these stressors are perceived to be frequent, recurring, severe, and exceeding the individually available coping resources, this can result in long-term maladaptive changes in these systems, namely chronic stress. 

Due to the capacity of symbolic thought in humans, the transition from acute to chronic stress may be associated with abstract stressors that emerge in adverse living and working conditions detrimental to individual well-being ([68]). In other words, chronic stress often involves permanent anxiety, exhaustion, or worries rather than an acute state of alarm due to the frequency of abstract stressors, inadequate adaptation, and relaxation failure ([70]). Therefore, chronic stress goes beyond typical daily hassles, namely short-term, frustrating everyday transactions, and is typically associated with a constant lack of different physical and emotional demands, such as loneliness, discrimination, or financial strain ([19]). Importantly, the prevalence and effects of chronic stress may differ between age cohorts and between groups that have different access to material and cultural resources. Whereas middle-aged adults apply coping mechanisms related to social networks more effectively than adolescents, older adults may face health problems and isolation ([3]). In addition, gender differences in chronic stress are frequently reported, typically suggesting that women have a higher risk of long-term, maladaptive stress (for a review, see [5]). Empirically, a representative German survey found that chronic stress was reported more often in women (13.90%) than in men (8.20%) and that there were significant differences between groups of low (17.30%) and high socioeconomic background (7.60%) ([27]). 

### 1.2. Chronic Stress and Cognitive Abilities

In general, cognitive abilities refer to components of mental functioning that involve encoding, manipulating, and storing information to learn in novel situations, acquire knowledge, and develop skills ([34]). While there is theoretical and empirical debate over the complex and multifaceted nature of intelligence (e.g., [21]), key concepts of most consensual intelligence theories are the ability to process new information (i.e., fluid intelligence) and storage of already acquired information (i.e., crystallized intelligence) (e.g., [51]). Fluid intelligence is a domain-general ability to solve abstract problems in novel situations, whereas crystallized intelligence comprises domain-specific knowledge and skills acquired through experience ([13]). Regarding the relation between fluid and crystallized intelligence, [14]’s ([14]) investment theory hypothesizes that fluid intelligence is primarily associated with biological factors. The development of fluid intelligence is frequently reported to be linked with cognitive maturation because there is a developmentally driven increase in abilities during the first twenty years of life, followed by a linear decline beginning in early adulthood ([63]). Crystallized intelligence, on the other hand, is typically reported to reach peak ability some years after that and plateaus during most of adulthood (e.g., [6]; [42]). Some authors even suggest that crystallized intelligence increases steadily until middle or late adulthood ([1]), highlighting its importance for adult intellectual performance ([2]; [28]). Regarding the development of these cognitive abilities, childhood and adolescence are typically marked by much more standardized learning environments than later adulthood, and crystallized intelligence is usually found to be malleable by learning experience ([44]).

Theories on the interrelation of cognitive abilities and different forms of stress typically hold that life events that are a perceived threat to homeostasis influence our well-being through stress-related neurobiological paths ([49]). The most important aspects of why theoretical frameworks highlight the connection between stress and cognitive abilities are the overlap in brain areas responsible for social and emotional information processing, the concepts of allostasis and allostatic load (i.e., long-term adaptation of the organism to stress), as well as the neuroplasticity that is reported in connection with acute and especially chronic stress ([49]). Theoretically, this is the basis for the “Glucocorticoid Cascade Hypothesis”, which postulates that through exposure to glucocorticoids, activity in components of the limbic network, such as the hippocampus, is altered, resulting in memory and learning impairments as well as information processing deficits (see [47]). Thus, chronic stress is generally held to have a negative and disruptive effect on important cognitive processes (e.g., [12]; [64]).

On a neurophysiological level, the functioning of the prefrontal cortex, which involves higher and more complex cognitive processing, the hippocampus, which involves memory, and the amygdala, which processes emotion- and stress-related behavioral and physiological information, is negatively affected by the increase in glucocorticoids ([35]; [49]). This overstimulation of the stress response system due to stress hormones is thought to have disruptive effects on the developing brain in children ([41]) and is associated with reduced hippocampal matter in adults ([48]). Overall, prolonged stressful life events shape our capacity and readiness for allostasis and adaptation, and they are associated with the psychological processes of apprehension, worry, and anxiety, which are known to have a negative effect on the performance in cognitive tasks (e.g., [46]). Thus, chronic stress can have a detrimental effect on basic and higher-order cognitive abilities through the long-term maladaptation of central components of the limbic network ([64]). Indeed, there is empirical evidence that links chronic stress with decreases in working memory ([24]), increases in mental health problems ([52]), and even the risk for dementia in old age ([76]).

### 1.3. Chronic Stress and Educational Attainment

While chronic stress appears to be a widespread phenomenon, there is evidence that certain groups in society are more vulnerable to the detrimental effects of chronic stress on cognitive abilities. Because of its transactional quality, how chronic stress develops is closely linked to the individually available coping resources ([22]; [47]). Such resources refer to proximal aspects of individual behavior as well as more structural resources linked to access to opportunities and privileges associated with our position in society ([32]; [49]). Socioeconomic status (SES) is a broad construct comprising several structural aspects of social stratification, such as education, occupation, and income ([23]). It can be argued that these endowed or acquired resources may lead to a differential, and possibly unequal, distribution of stress-related effects on cognitive abilities ([32]). Stressful life events that accumulate chronic stress are typically associated with a disadvantaged SES ([43]) because the limited availability of educational, social, and material resources may result in increased allostatic load ([49]). It is important to note that there is an ongoing debate about the proposed mechanisms of social causation and social selection (e.g., [26]). Empirically, SES has frequently been shown to be positively correlated with health and well-being as it is associated with the frequency and severity of stressful life events ([27]; [52]).

Educational attainment is a central component of SES linked to social participation, professional opportunities, and social mobility ([10]). It refers to a certain level reached in institutionalized academic contexts resulting in accumulated formal degrees or certificates. Typically, the education level is acquired during adolescence or early adulthood, but forms of further education in later life stages linked to the concept of lifelong learning have been on the rise in many countries, including Germany ([4]). The relationship between chronic stress and education is complex, as striving for academic achievement can also be stressful (for a review, see [57]). However, educational attainment in the sense of an acquired formal degree is typically regarded as negatively related to chronic stress ([36]), which is often attributed to the availability of coping resources to counter the negative effects of stressors ([54]). Empirically, it was found that education is, among others, positively associated with networks that foster support and social participation ([31]), which are argued to strengthen coping resources with regard to chronic stress ([50]).

### 1.4. The Current Study

The current study focused on the interrelation of chronic stress and cognitive abilities and how this interrelation varies in groups with different educational attainment. The individual links between chronic stress and cognitive abilities (for an overview, see [12]; [47]; [64]), chronic stress and education (e.g., [32]; [49]; [52]), as well as cognitive abilities and education (for an overview, see [42]; [44]; [61]) have been extensively studied. However, our knowledge about the interaction of these three elements in different age groups is limited. Therefore, we investigated the relationship between chronic stress, fluid and crystallized cognitive abilities, and educational attainment using cross-sectional data from a German panel study. Overall, we expected negative associations between chronic stress and cognitive abilities (e.g., [64]) (hypothesis 1). Because chronic stress is primarily held to impair memory ([18]), we reasoned that this negative association should be more pronounced for crystallized cognitive abilities (hypothesis 2). Finally, we compared groups that differed concerning their educational background. We expected that the negative association between chronic stress and cognitive abilities should be higher for lower education groups as these groups may lack access to resources for coping against stress ([50]) (hypothesis 3).

## 2. Materials and Methods

### 2.1. Sample

The current study used data from the adult cohort of the German National Educational Panel Study (NEPS SC6; [9]). NEPS SC6 had an annual data collection plan and drew on adults born between 1944 and 1986 living in Germany. Although the panel study was conceptualized as a representative cohort in Germany, panel attrition was selective regarding participants’ age, marital status, residency, household size, and education ([74]; [82]). Therefore, the NEPS SC6 data can be best described as large-scale secondary data from a heterogeneous, nationwide sample suitable to answer our research question on the interrelation of chronic stress, cognitive abilities, and education. In NEPS, all participants gave informed consent, and all ethical guidelines under German law for human subjects were followed by the Leibniz Institute for Education Trajectories (for more information on the ethical guidelines and data protection in NEPS, see [66]). Trained interviewers visited all individuals in their homes to administer a broad range of questionnaires that focused on different requirements, conditions, and consequences of education. In addition, competence tests were included on different ability domains in several survey waves ([9]). The current study focused on data from the sixth and seventh survey waves (data collection: August 2014 to March 2015 and August 2015 to March 2016, respectively; [53]), in which the data were collected by trained interviewers in the participants’ homes (i.e., self-report measures and computer-based assessment of cognitive abilities). Regarding inclusion criteria, we analyzed data from all individuals who participated in either one of these two survey waves, namely *N* = 10,416 individuals of NEPS SC6 (age range: *M* = 51.55, *SD* = 10.78, Min = 29, Max = 71; 50.8% female). A more detailed breakdown of the background variables can be found in the Appendix A. In addition to the variables of central importance to the present research question on the association of chronic stress and cognitive abilities, we report several sociodemographic aspects that are helpful for better understanding the overall sample composition.

### 2.2. Measures

Chronic Stress: For chronic stress, we drew on the Standard Stress Scale (SSS; [25]), which is a short self-report scale containing 11 items on various stressful life evaluations (e.g., social stress and anxieties, daily distress, worries about the future) rated on a 5-point Likert scale (see Appendix A). The instrument was developed specifically for the large-scale context of the NEPS, as it indicates a constant scale for different age groups covering a broad content range of mostly abstract and prolonged forms of abstract stress. All items were part of the interview conducted in the participants’ homes, which took, on average, 73 min to complete (face-to-face setting). Theoretically, the scale combines aspects of the job demand-control model ([33]) with the effort–reward imbalance model ([69]) to capture the mentally straining discrepancy between psychological demands or efforts and inadequate rewards, and it was validated regarding its’ explanatory effect on subjective health ([25]). The Standard Stress Scale was included in the seventh survey wave, resulting in information from *N* = 9347 cases. To account for the multifaceted structure of the instrument, we used a content-based parceling approach to model chronic stress as a unidimensional factor model, which resulted in a good model fit (CFI = 0.973, RMSEA = 0.083 [0.072; 0.096], SRMR = 0.021). We chose this approach because an item-based unidimensional model would not have yielded sufficient model fit (CFI = 0.805, RMSEA = 0.075 [0.073, 0.078], SRMR = 0.050). The scale has sufficient internal consistency, considering it covers various aspects of chronic stress (ω = 0.651).

Cognitive Abilities: One objective of NEPS SC6 was to collect data on a range of cognitive abilities of adults with different standardized tests ([80]). In the sixth survey wave of NEPS SC6, participants’ fluid and crystallized cognitive abilities were assessed with computer-based reasoning and receptive vocabulary tests, respectively. It should be noted that not all individuals in the net sample of the seventh survey wave also participated in the reasoning test (89.22%) and the receptive vocabulary test (89.43%). In total, information on cognitive abilities was available for *N* = 8827 cases. The participants were visited at home by a trained interviewer who administered the computer-based tests. Each test took the participants about 15 min, and the sequence of test administration was random at the beginning of the interview. Both tests were administered electronically on a tablet. More information on the test administration can be found in the official documentation ([45]).

Similar to established non-verbal tests, the reasoning test featured 12 different matrix-based items that participants should complete ([11]; [38]). All participants saw the same items, the test was not adaptive to the performance, and the participants had about 9 min to answer all items. Each item featured a sequence of eight geometric patterns, whereas the ninth missing pattern should be chosen from one of six options. An example item can be found in the online supplement (Appendix A). Internal consistency (ω = 0.730) and retest reliability (*r* = 0.68) were adequate and comparable to established tests ([38]). Convergent and divergent validity in adult samples was also sufficient ([38]). To reduce model complexity, we also used a parcel-based modeling approach by computing three parcels (based on four reasoning items each). 

Receptive vocabulary as an indicator of crystallized cognitive abilities was assessed with a shortened version of the German adaptation of the Peabody Picture Vocabulary Test that is suitable from the age of 13 onwards (PPVT-4; [39]). The shortened version was considered useful for the assessment in a large-scale context with limited administration time and a broad age range. It featured 89 items, and the participants had to select one out of four pictures that matched the word presented by a standardized audio track. Internal consistency was good (ω = 0.868), although the probability of correctly solving the items was generally high (*M* = 0.85, *SD* = 0.16, Min = 0.34, Max = 0.99). We again chose a parcel-based approach to reduce model complexity by computing five parcels (with items randomly assigned to the parcels to reduce order effects). A correlated-factors model, including both reasoning and vocabulary, had a good model fit (CFI = 0.978, RMSEA = 0.060 [0.056; 0.064], SRMR = 0.029).

Educational Attainment and Control Variables: In the current study, educational attainment is used as an indicator of endowed or acquired social and individual resources. We used the International Standard Classification of Education (ISCED-97; [77]), which was adapted for Germany ([67]). More specifically, we grouped subjects into three groups of educational attainment (elementary level, secondary level, and tertiary level; see Appendix A for details). In total, *n* = 401 subjects were in the elementary-level education group, *n* = 4253 subjects were in the secondary-level education group, and *n* = 4629 subjects were in the tertiary-level education group. 

Regarding control variables, gender differences are frequently reported for chronic stress ([5]; [7]), so we included a dichotomous variable of the participants’ gender (self-report). In the models, we also included the participants’ age as a control variable because it is related to cohort effects of education as well as to fluid cognitive abilities ([6]; [42]). We used a household-size adjusted indicator of equivalized monthly net income to control for financial resources ([55]). Other control variables that might influence the experience of chronic stress were dichotomous indicators of marital status (for a critical discussion on coping and social support, see [71]; [73]) and retirement status ([79]), as well as household language as a dichotomous indicator of potential acculturative stress ([62]).

### 2.3. Data Analysis

All analyses were conducted using R version 4.3.1 ([59]). For our latent variable analyses, we mainly relied on the package *lavaan* version 0.6-16 ([60]). All data are available for researchers via www.neps-data.de. To make the present analyses transparent and reproducible, we provide all Appendix A (i.e., R syntax to reproduce the results, and Appendix A) online via osf.io/vcfds (accessed on 16 January 2025).

Missing Data: Due to the survey design, different types of missing data occurred. There were participants who either answered only the self-report type data (i.e., chronic stress; *n* = 1589), participants who had only information on the ability assessments (i.e., reasoning and vocabulary; *n* = 1069), or participants who had information on both (*n* = 7758). For the present analyses, we used data from all participants, resulting in the final *N* of 10,416. Apart from this design-related missingness, there are also missing values for single items. Accordingly, for our latent variable analyses, we use *full information maximum likelihood* (fiml) to deal with missingness. Since exogen variables cannot be imputed using fiml, we impute missing values for our control variables using the k-nearest neighbor algorithm implemented in the package *VIM* version 6.2.2 ([37]). In addition, we report two kinds of sensitivity analyses to investigate the robustness of our findings: First, we report our analyses using the subsample of subjects who have information for both chronic stress and cognitive ability. Second, as the elementary education group is significantly smaller than the two other groups, we use a propensity score matching approach based on the k-nearest neighbor algorithm implemented in the package *MatchIt* version 4.6.0 ([30]) to select subgroups within the other educational groups that match with regard to our control variables.

Latent Variable Analyses: To investigate the influences of age and education on the relationship between chronic stress and cognitive abilities, we used a latent variable modeling approach. First, since education is a categorical variable, we used multigroup confirmatory factor analysis to illustrate potential differences between the educational groups after establishing scalar measurement invariance ([58]; [78]). Since our sample size is comparatively big and the models under investigation complex, we based our inferences on measurement invariance on fit indices that are independent of sample size and model complexity ([16]), following the cut-off criteria proposed by [15] ([15]): ΔCFI > 0.01, ΔRMSEA > 0.015, and ΔSRMR > 0.03.

## 3. Results

In Table 1, we report descriptive statistics and correlations of our continuous demographic variables and the parcels and overall sum scores of chronic stress and cognitive abilities (see also Appendix A for a more detailed overview of the descriptive statistics separated by educational group). Notably, the vocabulary test shows high scores, indicating a potential ceiling effect. However, the overall correlational pattern is consistent with our expectations. On a manifest level, both sum scores of reasoning and vocabulary are negatively correlated with chronic stress. Interestingly, the different facets of the chronic stress scale show differential correlations with both reasoning and vocabulary. For example, the work-related facet seems to be only related to vocabulary and not to reasoning, and also, the exhaustion-related facets correlate only very weakly (if at all) with reasoning. 

Figure 1 shows the basic model of the relationship between chronic stress and cognitive abilities. The model includes age, gender, income, retirement status, household language, and marital status as control variables (see Appendix A for the basic model without control variables and Appendix A for the relation between the latent factors and the control variables). In line with hypothesis 1, we found a small but significant negative relation between chronic stress and cognitive abilities. To further compare the magnitude of both correlation coefficients, we computed the model with the correlations fixed to equality. This restriction did not lead to a meaningful deterioration in model fit (CFI = 0.970, RMSEA = 0.043 [0.042; 0.045], SRMR = 0.032). Accordingly, hypothesis 2 was not supported. A similar pattern emerged for the sensitivity analyses using the reduced sample with participants who completed both assessments of chronic stress and cognitive abilities. Not only did the models yield comparable fit (see Appendix A), but also the magnitude of estimated effect sizes were similar with a statistically significant latent correlation of −0.07 between chronic stress and fluid abilities and −0.08 between chronic stress and crystallized abilities, respectively, indicating no bias due to our treatment of missing data (see Appendix A).

Next, we used the basic model described above to include in a multigroup confirmatory factor analysis. In the model, all participants who provided information on their educational attainment were included (*n* = 9283). We were able to establish scalar measurement invariance (see Table 2). 

Overall, no clear pattern emerged concerning the correlations between chronic stress and cognitive abilities within these three groups. Effects were comparatively small in all groups (ranging from −0.056 to −0.099). In Table 3, we report latent covariances and factor means of the three groups. Constraining the correlations between the latent variables to equality between groups did not lead to a meaningful deterioration in model fit (CFI = 0.959, RMSEA = 0.046, SRMR = 0.040), indicating no significant differences between the groups. Similarly, fixing the latent means to equality between groups also led to no meaningful deterioration in model fit (CFI = 0.959, RMSEA = 0.046, SRMR = 0.038). As a final sensitivity analysis, we applied propensity score matching to compare cases between the education groups (considering all sociodemographic factors but income) to assess whether the lack of effects was due to the unevenly distributed sample sizes of the individual groups. The overall pattern of effects was not different from the overall model (Appendix A), indicating no bias due to unequal group sizes or group composition. Taken together, this means that the group differences in estimates reported in Table 3 should be interpreted with caution. Although for the relation between reasoning and chronic stress, on a descriptive level, the latent correlation is the largest in the elementary-level education group and smallest in the tertiary-level education group, these differences cannot be interpreted in support of hypothesis 3. 

## 4. Discussion

In the current study, we used a large cross-sectional German dataset to investigate the relationship between chronic stress, cognitive abilities, and educational attainment in adults. Consistent with our expectations, we found small, negative associations between fluid and crystallized abilities and chronic stress that remained significant even after controlling for demographic information that might influence stress and/or cognitive abilities. However, there was no evidence that chronic stress affects one type of cognitive ability more than the other, as there was no difference in the magnitude of the relationship between stress and fluid or crystallized abilities, respectively. Moreover, there was also no clear evidence for differential effects of chronic stress and cognitive abilities between education groups, with roughly the same effect size in all educational groups. However, the descriptive differences, in combination with the comparably small sample size of the low-education group, suggest that significant differences might be found in more balanced datasets.

### 4.1. Theoretical and Practical Implications

Although a causal interpretation is not possible due to the cross-sectional data, our results mirror previous empirical findings (previous reviews did not focus on education; see [43]; [47]; [64]) showing that both fluid and crystallized abilities are negatively associated with chronic stress. In other words, people with lower cognitive abilities also reported higher levels of emotional strain due to prolonged and abstract forms of anxiety, exhaustion, or worries. This finding supports the idea that limited material or cultural resources are related to both our cognitive abilities and our capacity for allostasis ([32]; [49]). It should be noted that this is probably a bidirectional process, namely that on the one hand, lower cognitive abilities result in fewer coping opportunities or less efficient coping strategies when facing stressful events, while on the other hand, chronic stress may harm neuronal processes and therefore cognitive abilities in the long run. 

Contrary to our expectations, fluid and crystallized abilities are equally related to chronic stress. While the descriptive correlations suggest differential effects for both cognitive abilities, the models indicated negative effects that were similar in magnitude, especially after controlling for demographic information. It seems likely that both cognitive abilities are influenced in complex ways by these work-related or social worries. It could be that these worries limited the participants’ capacity to fully engage in the tests, namely by a distraction of the attention system that helps to process new information ([75]). While such effects were primarily shown for burnout and depression patients ([47]; [64]), this may also impair attention in non-clinical samples. Finally, motivational aspects could also contribute to how the participants performed in both tests. Studies that focus on resilience (i.e., the ability to maintain homeostasis when exposed to stressful events) typically show that chronic stress has a negative effect on motivation and endurance in various assessments (for an overview, see [17]).

Surprisingly, we did not find clear evidence of systematic differences between education levels. The broad age range of the cross-sectional sample and the comparably small sample size of the low-education group might have obscured potential differences. More specifically, the overall prevalence of certain education levels, as well as their relation to cultural and material resources (i.e., returns), such as job opportunities and social prestige, have changed in the last decades in Germany. In short, younger generations tend to acquire higher qualifications, while returns from lower qualifications have diminished over time (e.g., [8]). In addition, while educational degrees are central to social participation and the development of knowledge and skills, a protective effect that could be expected regarding the interrelation between chronic stress and cognitive abilities ([50]) might also depend on how long ago the degree was acquired, although positive effects on mental health seem to be rather long-lasting ([36]). Finally, it should be mentioned that educational degrees can only be regarded as a proxy for more direct coping mechanisms that might also be associated with work conditions ([29]).

Because the current findings are based on cross-sectional data that preclude causal inferences, we must be careful when deriving specific practical implications. In general, the study highlights the critical role of chronic stress in cognitive abilities. While we cannot comment on whether chronic stress is the reason for or the result of interindividual differences in fluid and crystallized cognitive abilities, the findings support the notion that chronic stress is an indicator of potential mental health issues. In addition, the results—at least at a descriptive level—hint at a positive role of education in the interrelation between chronic stress and cognitive abilities (especially for reasoning), possibly due to cultural resources such as coping strategies (see also the literature on resilience, e.g., [65]). Therefore, further education could be relevant for enhancing and updating (older) adults’ knowledge and skill levels ([4]), ultimately leading to reduced stress levels as the individual improves their skills in dealing with everyday challenges. In addition, further education might reduce social stressors such as loneliness and isolation due to increased social participation, although the research is still scarce (for a review, see [40]). However, it should be noted that striving for higher levels of educational attainment is probably not helpful when already experiencing elevated levels of chronic stress due to the expected (but possibly short-term) aggravation of stress-inducing demands in (semi-)institutional learning settings ([57]).

### 4.2. Limitations and Future Directions

Several limitations may have influenced the results of the present study. Firstly, all instruments measuring stress and stress-related variables were based on self-report questionnaires. In other words, the items might have also captured personality aspects, such as neuroticism, which has equivalent associations with cognitive abilities (for a meta-analysis, see [72]). While self-report measures are often biased by social desirability, they should capture the subjective perspective, which is relevant for cognitive appraisal in chronic stress (e.g., [43]). Also, while the instrument was specifically designed to capture a broad range of different stressors, the limited number of items per domain does not allow for a more detailed consideration of the individual domains of the experience of chronic stress. However, on a descriptive level, our results provide an initial indication that domain-specific effects might occur, as systematic patterns in the first-order correlations between cognitive abilities and the different dimensions of chronic stress can be observed. Accordingly, future studies should address the question of whether the experience of stress in certain areas of life has a differential influence on the relationship with cognitive abilities. At the same time, it seems sensible to investigate whether the negative effects of stressful experiences in a particular domain can be compensated for by other areas of life.

Second, chronic stress and cognitive abilities were not measured in the same survey wave. Although cognitive abilities should be comparatively stable during the short time frame of approximately 1 year ([28]), the assessment of fluid and crystallized abilities took place before the assessment of chronic stress. Per definition, chronic stress is a condition that persists over a longer period, rendering bias due to the delayed assessment unlikely. However, due to the cross-sectional design, no aspects of the length of exposure to stress-related events could be tested. This could have further obscured differences between age cohorts due to the cumulative effects of chronic stress on allostatic load ([48]). With the data, the present results are to be interpreted as bidirectional associations. Ideally, future research should adopt a longitudinal perspective on both chronic stress and cognitive abilities, allowing us to model changes in both aspects simultaneously. This approach might also help to depict the cumulative effects of prolonged stress ([48]). Some stressors may have been present for a long time, and thus, some consequences of chronic stress could manifest early in life. This is because, throughout an individual’s lifespan, different vulnerabilities and protective factors accumulate ([49]). Therefore, cohort differences, age-related changes in the variance of cognitive-test scores, age differences in population representativeness, age-related differences in measurement reliability, and other methodological factors, such as selective attrition ([82]), are possible confounds in cross-sectional data, making a longitudinal perspective necessary.

## 5. Conclusions

The current study investigated the interrelation between chronic stress, cognitive abilities, and educational attainment using cross-sectional data from a German panel study. We found small negative effects of chronic stress on fluid and crystallized abilities, respectively. In addition, we found no significant differences across educational groups, although our sample size was sufficient to detect even small variations. Overall, our study does not address longitudinal trends or potential cumulative effects of chronic stress on cognitive abilities, but the results are still relevant to our understanding of how acquired educational attainment as a cultural resource might contribute to how we are exposed to or how well we adapt to stressful life events that can impair basic cognitive processes.

## Figures and Tables

**Figure 1 jintelligence-13-00013-f001:**
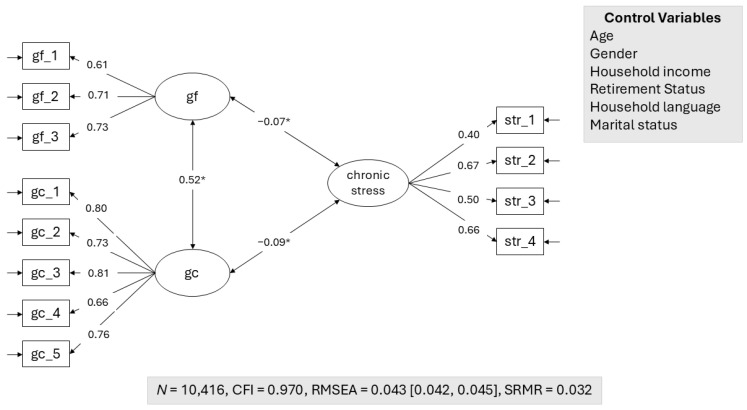
Relation between chronic stress and cognitive abilities; * *p* < 0.05.

**Table 1 jintelligence-13-00013-t001:** Descriptive statistics of chronic stress indices, reasoning, and vocabulary.

		Descriptives							Correlations															
		*N*	*M*	*SD*	Min	Max	skew	kurt	1	2	3	4	5	6	7	8	9	10	11	**12**	**13**	**14**	**15**	**16**
Demographics																								
1	Age	9347	51.55	10.78	29.00	71.00	−0.27	−0.70																
2	Income	7057	2171.84	977.06	0.00	5500.00	1.13	1.54	**0.03**															
Chronic Stress																								
3	Sum Score	9277	27.07	5.07	11.00	54.00	0.41	0.57	**−0.07**	**−0.23**														
4	Work	9347	2.81	0.56	1.00	5.00	−0.07	0.28	**−0.12**	**−0.13**	**0.59**													
5	Social	9346	1.93	0.59	1.00	5.00	0.72	0.87	**−0.04**	**−0.19**	**0.73**	**0.31**												
6	Exhaustion	9346	2.93	0.76	1.00	5.00	0.23	−0.11	**−0.10**	**−0.11**	**0.74**	**0.20**	**0.31**											
7	Future	9342	2.04	0.81	1.00	5.00	0.76	0.53	**0.08**	**−0.23**	**0.70**	**0.21**	**0.44**	**0.36**										
Reasoning																								
8	Sum Score	8798	8.20	2.76	0.00	12.00	−0.70	−0.24	**−0.39**	**0.20**	**−0.06**	0.01	**−0.05**	−0.02	**−0.13**									
9	Parcel 1	8787	0.71	0.26	0.00	1.00	−0.76	0.01	**−0.27**	**0.14**	**−0.05**	0.01	**−0.03**	**−0.03**	**−0.09**	**0.74**								
10	Parcel 2	8763	0.67	0.28	0.00	1.00	−0.59	−0.43	**−0.31**	**0.17**	**−0.05**	0.00	**−0.04**	0.00	**−0.10**	**0.80**	**0.41**							
11	Parcel 3	8751	0.72	0.29	0.00	1.00	−0.85	−0.19	**−0.31**	**0.16**	**−0.04**	0.02	**−0.04**	0.00	**−0.10**	**0.82**	**0.45**	**0.52**						
Vocabulary																								
12	Sum Score	8818	73.62	9.65	0.00	89.00	−1.82	6.00	−0.02	**0.29**	**−0.12**	**−0.06**	**−0.10**	**−0.05**	**−0.14**	**0.43**	**0.30**	**0.36**	**0.33**					
13	Parcel 1	8815	0.92	0.10	0.00	1.00	−2.39	8.47	0.00	**0.20**	**−0.08**	**−0.04**	**−0.07**	**−0.03**	**−0.10**	**0.34**	**0.24**	**0.29**	**0.26**	**0.79**				
14	Parcel 2	8811	0.90	0.10	0.17	1.00	−1.72	5.16	**0.08**	**0.17**	**−0.06**	**−0.04**	**−0.06**	−0.02	**−0.07**	**0.28**	**0.20**	**0.24**	**0.21**	**0.73**	**0.62**			
15	Parcel 3	8812	0.80	0.15	0.00	1.00	−0.95	0.90	**0.10**	**0.26**	**−0.11**	**−0.07**	**−0.09**	**−0.05**	**−0.11**	**0.33**	**0.24**	**0.28**	**0.26**	**0.80**	**0.64**	**0.57**		
16	Parcel 4	8814	0.77	0.12	0.00	1.00	−0.69	0.91	**−0.07**	**0.26**	**−0.09**	**−0.05**	**−0.08**	−0.02	**−0.10**	**0.37**	**0.27**	**0.32**	**0.31**	**0.69**	**0.52**	**0.44**	**0.58**	
17	Parcel 5	8817	0.86	0.11	0.00	1.00	−1.40	3.61	**0.13**	**0.21**	**−0.10**	**−0.05**	**−0.09**	**−0.05**	**−0.10**	**0.26**	**0.20**	**0.23**	**0.19**	**0.74**	**0.61**	**0.56**	**0.61**	**0.48**

*Note.* Pearson correlation coefficients based on sample sizes ranging between 5856 and 9347; significant correlations in bold (*p* < 0.05).

**Table 2 jintelligence-13-00013-t002:** Measurement invariance.

	CFI	ΔCFI	RMSEA	ΔRMSEA	SRMR	ΔSRMR
Configural	0.968		0.043		0.033	
Metric	0.966	0.002	0.043	<0.001	0.035	0.002
Scalar	0.960	0.006	0.045	0.002	0.037	0.002

*Note*. CFI = Comparative Fit Index, RMSEA = Root Mean Square Error of Approximation, SRMR = Standardized Root Mean Square Residual, Δ indicates the change in the respective fit index between nested models. Changes are calculated as the fit index of the less constrained model subtracted from that of the more constrained model.

**Table 3 jintelligence-13-00013-t003:** Latent covariances and factor means.

		Educational Level					
		Elementary		Secondary ^a^		Tertiary	
		est.	*p*	est.	*p*	est.	*p*
Latent Covariances							
	gf/gc	0.594	<.001	0.500	<.001	0.479	<.001
	gf/stress	−0.164	0.075	−0.058	0.025	−0.042	0.098
	gc/stress	−0.068	0.390	−0.079	0.001	−0.067	0.002
Factor Means							
	gf	−0.649	0.088	0	NA	0.129	0.470
	gc	−0.797	0.022	0	NA	0.821	<.001
	stress	0.500	0.199	0	NA	−0.829	<.001

*Note*. ^a^ Secondary educational level was used as a reference group with factor means fixed to zero; NA—not available.

## Data Availability

This paper uses data from the National Educational Panel Study (NEPS; see [9]) Starting Cohort Adults ([53]). From 2008 to 2013, NEPS data were collected as part of the Framework Program for the Promotion of Empirical Educational Research funded by the German Federal Ministry of Education and Research (BMBF). As of 2014, NEPS has been carried out by the Leibniz Institute for Educational Trajectories (LIfBi, Bamberg) in cooperation with a nationwide network. All data are available for researchers via http://www.neps-data.de (accessed 16 January 2025).

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
