# Peer review of "The Mind Under Pressure: What Roles Does Education Play in the Relationship Between Chronic Stress and Cognitive Ability?"

_jintelligence, 2025, doi:10.3390/jintelligence13020013_

Round 1
Reviewer 1 Report
Comments and Suggestions for Authors
Thank you for being able to read this manuscript.
The abstract does not provide information about the method, participants or instruments (all within the method) and should include a brief introduction, objectives, method, results and conclusions without these statements having to appear, as the format of the journal must be respected.
In the introduction there are many references together, I do not think it is necessary, and in addition many of them are more than 5 years old, they should be removed, unless theoretically very relevant.
I don't quite understand the procedure, nor if there was an ethical approval, nor how they went to the homes of these people, furthermore, there is no table of socio-demographic data that explains what specific characteristics these people had, we don't know if they had a disability, or a specific health problem. There are no inclusion and exclusion criteria. So everything else refers to age but even then there should be age ranges. I encourage you to complete the information.
Author Response
Comment 1: The abstract does not provide information about the method, participants or instruments (all within the method) and should include a brief introduction, objectives, method, results and conclusions without these statements having to appear, as the format of the journal must be respected.
Response: Thank you for the comment, we have revised the abstract according to the journal guidelines and added the relevant information where necessary. In the revised abstract (p. 1), we provide more information about the method, participants, and instruments within the 200-word limit.
Comment 2: In the introduction there are many references together, I do not think it is necessary, and in addition many of them are more than 5 years old, they should be removed, unless theoretically very relevant.
Response: We have updated the introduction to focus only on studies that are theoretically and empirically necessary. While some studies do not fall within a 5-year limit (e.g., the review on cognitive functioning and chronic stress by Sandi 2013), we made sure that these are still the most recent publications focusing on this particular topic (p. 1). To the best of our knowledge, the findings now cited in the revised manuscript reflect the relevant findings from the respective research areas.
Comment 3: I don't quite understand the procedure, nor if there was an ethical approval, nor how they went to the homes of these people, furthermore, there is no table of socio-demographic data that explains what specific characteristics these people had, we don't know if they had a disability, or a specific health problem. There are no inclusion and exclusion criteria. So everything else refers to age but even then there should be age ranges. I encourage you to complete the information.
Response: We recognize the need to improve information about the sample and the overall process. We used data from a nationwide survey in Germany (German National Educational Panel Study) that is freely available to the scientific community. This was considered a useful approach due to the heterogeneous and representative sample structure. Regarding ethical approval, although the procedures are different from, for example, experimental or clinical research, the ethical standards are reviewed by the Leibniz Institute for Educational Trajectories and are in accordance with the German law on collecting data from individuals. An updated ethical approval statement has been submitted to the editorial team. The procedure is described in more detail in the updated version: Briefly, trained interviewers were sent to participants' homes to conduct an interview (on average 73 minutes) and computer-based ability tests (on average 30 minutes). In the manuscript, we revised the section on sample description, and we now provide additional information on sample characteristics in the online supplement (pp. 4-5). Due to the extensive data collection plan of NEPS SC6 with a focus on education, specific health information is unfortunately not available in the data. However, we added information on general impairments/disabilities officially recognized in Germany, and ADHD diagnoses specifically in the sample description in the online supplement (Table S1).
Reviewer 2 Report
Comments and Suggestions for Authors
The manuscript "mind under pressure" is a secondary data analysis of cross sectional data of a representative sample. The paper is well written and I love that the authors comply with open science practices.
I only have a few minor comments.
Regarding hypothesis 3, I suggest to use propensity matching (match it package in R) so that the ca 400 in the lowest educational group are matched with the same number of participants having secondary and tertiary level - by the demographic factors (age, gender, marital status, retirement status, household language, for obvious reasons not by income)
This may strengthen the conclusion for hypothesis 3
regarding the argument of crystalline intelligence over the lifespan, a very good resource is Hartshore and Germine, 2015 https://journals.sagepub.com/doi/full/10.1177/0956797614567339
minor:
line 10: chronic stress - stress missing
line 169 groups - s missing
line 318 -.07 <- line break where there should be none
Author Response
Comment 1: Regarding hypothesis 3, I suggest to use propensity matching (match it package in R) so that the ca 400 in the lowest educational group are matched with the same number of participants having secondary and tertiary level - by the demographic factors (age, gender, marital status, retirement status, household language, for obvious reasons not by income). This may strengthen the conclusion for hypothesis 3.
Response: Thank you for the suggestion, we added the analyses based on matched samples as a second sensitivity analysis. We briefly report results in the results section of the updated manuscript (p. 9) and provide additional information in the revised version of the supplement (Table S7; Table S8).
Comment 2: Regarding the argument of crystalline intelligence over the lifespan, a very good resource is Hartshore and Germine, 2015.
Response: Thank you for suggesting this interesting article. The study has an impressive database and the different patterns of cognitive functioning in different age groups are remarkable. We have added the reference to the chronic stress and cognitive ability section (p. 3) as well as the discussion (p. 11).
Round 2
Reviewer 1 Report
Comments and Suggestions for Authors
I understand that the authors have made an effort to improve the manuscript but there are aspects that are not clear, one is still the procedure, there is no section that explains in detail how they went to people's houses and gave them a survey, did they all accept it, etc.
On the other hand, I have seen in S1 that disability was among the socio-demographic characteristics, but I see nothing about it in the article, how can it be justified that certain variables measured are not even mentioned? Finally, within the socio-demographic variables, no attention is paid to cultural diversity, place of origin, religion and much less sexual orientation, how can this be justified in a population-based survey, because we know that many of these variables are affecting intelligence.
Greetings
Author Response
Comment 1: I understand that the authors have made an effort to improve the manuscript but there are aspects that are not clear, one is still the procedure, there is no section that explains in detail how they went to people's houses and gave them a survey, did they all accept it, etc.
Response 1: Thank you for the comment, we understand that it is crucial to understand the data at hand, especially with secondary data analysis. The German National Educational Panel Study is a large-scale nationwide survey with an annual data collection plan. Trained interviewers administer questionnaires (i.e., self-report data) as well as competence tests (i.e., direct measures) in the homes of the participants. The survey focuses on the educational context, which is why data on a range of different topics is collected in a typically short and general manner (as is usual in comparable panel studies). NEPS comprises several sub-studies focusing on specific sociodemographic groups; the current paper used data from an adult cohort (Starting Cohort 6; “NEPS SC6”) featuring several thousand adults living in Germany. We have added more information on the scope of NEPS in the section “2. Materials and Methods: Sample”.
“Therefore, the NEPS SC6 data can be best described as large-scale secondary data from a heterogeneous, nationwide sample suitable to answer our research question on the interrelation of chronic stress, cognitive abilities, and education. In NEPS, all participants gave informed consent, and all ethical guidelines under German law for human subjects were followed by the Leibniz Institute for Education Trajectories (for more information on the ethical guidelines and data protection in NEPS, see Schier et al. 2019). Trained interviewers visited all individuals in their homes to administer a broad range of questionnaires that focused on different requirements, conditions, and consequences of education. In addition, competence tests were included on different ability domains in several survey waves (Blossfeld and Roßbach 2019). […] In addition to the variables of central importance to the present research question on the association of chronic stress and cognitive abilities, we report several sociodemographic aspects helpful for better understanding the overall sample composition.” (manuscript pp. 4–5)
We want to draw your attention to the fact that all data from the German National Educational Panel Study, as well as its documentation, are freely available for the research community (in the case of NEPS SC6, see https://www.neps-data.de/Data-Center/Data-and-Documentation/Start-Cohort-Adults/Documentation). Therefore, we chose to include only information within the limitations of a typical scientific manuscript. Exploring the extensive official documentation provided by NEPS, a more detailed overview is possible, even regarding further analyses to check the validity of our results. Following an open science approach, ensuring the reproducibility of scientific results, we provide the syntax for all reported calculations.
Finally, we invite you to look at the following list of citations that we also refer to in the paper that give a more nuanced look into the data collection process.
For the aims and scope of NEPS: Blossfeld, Hans-Peter, Jutta von Maurice, and Thorsten Schneider. 2019. “The National Educational Panel Study: Need, main features, and research potential.” In Education as a lifelong process: The German National Educational Panel Study (NEPS). Edited by Hans-Peter Blossfeld and Hans-Günther Roßbach, 1–16. Springer VS. https://doi.org/10.1007/978-3-658-23162-0_1.
Data collection process in the adult cohort of NEPS (only in German): Malina, Aneta, Angelika Steinwede, Doris Hess, Frédéric Turri, Volker Aust, and Martin Kleudgen. 2015. “Methodenbericht NEPS-Startkohorte 6 Haupterhebung 2014/2015 B97 [Method Report NEPS Starting Cohort 6 Main Survey 2014/2015 B97].” infas Institut für angewandte Sozialwissenschaft GmbH. https://www.neps-data.de/Portals/0/NEPS/Datenzentrum/Forschungsdaten/SC6/7-0-0/Methodenbericht_SC6_W7_B97.pdf.
Survey content (questionnaires; competence tests not included): Leibniz Institute for Educational Trajectories. Starting Cohort 6: Adults (SC6). Waves 4 and 5 questionnaires (SUF version 6.0.0). Leibniz Institute for Educational Trajectories. https://www.neps-data.de/Portals/0/NEPS/Datenzentrum/Forschungsdaten/SC6/6-0-0/NEPS_SC6_SurveyInstruments_SUF_6-0-0_w4-5_en.pdf.
Ethical and data protection guidelines (applies to all NEPS sub-studies): Schier, Antonia, Meike Bender, Tobias Koberg, Brigitte Bogensperger, Sonja Gruner, David Schiller, Jutta von Maurice, and Henriette Engelhardt-Wölfler. 2019. “Data protection issues in the National Educational Panel Study.” In Education as a lifelong process: The German National Educational Panel Study (NEPS). Edited by Hans-Peter Blossfeld and Hans-Günther Roßbach, 347–360. Springer VS. https://doi.org/10.1007/978-3-658-23162-0_18.
Comment 2: On the other hand, I have seen in S1 that disability was among the socio-demographic characteristics, but I see nothing about it in the article, how can it be justified that certain variables measured are not even mentioned?
Response 2: Yes, you are right, disability was not further explored. We included this information in the revised version of the manuscript to provide additional information on the sample characteristics. Because of the large-scale approach of NEPS, a broad range of information on participants’ education was collected. In our current study, we only focus on the association between chronic stress, cognitive abilities, and educational achievement. While we believe that other sociodemographic aspects should be important to tap into the underlying mechanisms (see also Response 3), it should be noted that most information in NEPS is based on very general questionnaires or assessments. Regarding disability, for example, it was only asked whether the participants had a certain degree of disability recognized under German law (binary variable; “Versorgungsmedizin-Verordnung”). We included this information in the section “2. Materials and Methods: Sample” and Table S1 in the supplement.
Comment 3: Finally, within the socio-demographic variables, no attention is paid to cultural diversity, place of origin, religion and much less sexual orientation, how can this be justified in a population-based survey, because we know that many of these variables are affecting intelligence.
Response 3: Thank you for pointing out the importance of variables that could further explain the association between chronic stress, cognitive abilities, and educational achievement. Cognitive abilities have many correlates that have been studied extensively in the past decades – ranging from specific personality traits (e.g., Von Stumm and Ackerman 2013) to broad life outcomes (e.g., Kuncel et al. 2004). While each of these topics is certainly important, in this paper we focus primarily on factors particularly relevant when considering the relationship between cognitive ability, chronic stress, and education. In this context, it is essential to note that many of the factors you bring up in your commentary are still the subject of current debates in intelligence research, which are beyond the scope of the current manuscript; see, for example, the debate surrounding the measurement and modeling of cultural differences in intelligence performance (for an overview of methodological challenges and shortcomings of studies focusing on the German context, see Schroeders and Wilhelm 2020). However, we understand that, for example, religiosity (for a meta-analysis, see Zuckerman et al. 2013) and sexuality (for homosexuality, see Kanazawa 2012) may be important in understanding societal variations in cognitive abilities. Information on these aspects could be available to a certain degree in NEPS SC6, although possibly in a very general way (see also Response 2). In addition, the sample size of such groups could be rather small for more elaborate analyses (see Table S1). Overall, the current study focused on chronic stress, cognitive abilities, and educational achievement at the population level. While you raised important questions regarding other sociodemographic aspects associated to varying degrees with these factors, we think future research should investigate these mechanisms in more detail.
References
Kanazawa, Satoshi. 2012. “Intelligence and homosexuality.” Journal of Biosocial Science 44 (5): 595–623. https://doi.org/10.1017/S0021932011000769.
Kuncel, Nathan R., Sarah A. Hezlett, and Deniz S. Ones. 2004. “Academic performance, career potential, creativity, and job performance: Can one construct predict them all?” Journal of Personality and Social Psychology 86 (1): 148–161. https://doi.org/10.1037/0022-3514.86.1.148.
Schroeders, Ulrich and Oliver Wilhelm. 2020. “Es gibt drei Arten von Lügen: Lügen, verdammte Lügen und Statistiken – Kommentar zu Klauk (2019) [There are three types of lies: Lies, damned lies, and statistics – Commentary to Klauk (2019)].“ Wirtschaftspsychologie 2: 45–48.
Von Stumm, Sophie, and Phillip L. Ackerman. 2013. „Investment and intellect: A review and meta-analysis.” Psychological Bulletin 139 (4): 841–869. https://doi.org/10.1037/a0030746.
Zuckerman, Miron, Jordan Silberman, and Judith A. Hall. 2013. “The relation between intelligence and religiosity: A meta-analysis and some proposed explanations.” Personality and Social Psychology Review 17 (4): 325–354. https://doi.org/10.1177/1088868313497266.
Reviewer 2 Report
Comments and Suggestions for Authors
Thank you very much for the revision. I have no further issues, and congratulate the authors for their great work (not least the very informative supplementary materials).
Author Response
Thank you for your comments. We believe that your input helped in improving the quality of the manuscript. Please note that based on another reviewer's comments, we added minor information on the sample structure in the section "2. Materials and Methods: Sample".
Round 3
Reviewer 1 Report
Comments and Suggestions for Authors
The authors have made a major effort to improve the manuscript and I think they have explained the doubts in their comments. Regards